## [Peer Review File · Nature Communications]

Reviewers' comments:

Reviewer #1 (Remarks to the Author):

The manuscript describes coherent microwave control of the electron spin of single SiV- in diamond. This topic is of interest because of the good optical quality of the fluorescence compared to the NV-. The results are novel, being an extension of the coherent population trapping work from 2014. The data quality is good and the presentation is very good (apart from the comments below). I recommend publication once the three questions below are resolved.

1. At the end of page 5 (line 123) measurements of $T_{1,orbital}$ are mentioned, but there is no mention in the manuscript of how these data were measured, or any presentation of these data. Please introduce these measurements in the text and show the data. If this refers to data from a previous paper then reference this clearly.
2. Why is the value of T_2^* longer than that in the 2014 CPT papers? Please explain this in the paper.
3. The hyperfine is given as 70MHz, but in Rogers et al (PRL 113, 263602 (2014)) it is given as 35MHz. Presumably this is a different convention? Please explain in the paper.

Further minor points:

1. It could be good to mention in the title that a single SiV- is being used
2. On line 126 there is $1/T_2$ but presumably this should be $1/T_2^*$
3. On lines 130, 136 and in the supplementary info there is $1/2T_{1,orbital}$ and $1/2T_{1,spin}$. It would be clearer to put brackets around the denominator.
4. The 2 panes of figure 5 are too close to each other so the axes labels are almost touching.

Reviewer #2 (Remarks to the Author):

In this manuscript, the silicon-vacancy (SiV) center in diamond is investigated. The negatively charged SiV center emits around 80 % of its photons into the zero phonon line, with optical properties characterized by spectral stability and narrow inhomogeneous distribution in bulk diamond. In this manuscript, authors report the first realization of coherent control of the SiV center electron spin. They perform optically detected magnetic resonance (ODMR) by tuning the frequency of a microwave field into resonance with the Zeeman splitting between two electron spin states of a single SiV center.

From the excellent optical properties, SiV center is significantly interesting. The electron spin states and its magnetic field dependences has been already reported. All optical initialization, readout, and coherent preparation of single SiV spin has also reported. I agree that the observation by ODMR is important from the view point of coherent control of spin by microwave.

I have several questions and comments as follows.

(1) In previous study referred in [20], the coherence time (T_2^*) is reported to be 35 ns. I would like to ask the reason why $T_2^* = 115$ ns in the present study is much longer than it. The authors conclude the T_2^* is limited by T_{1spin} in the present study. On the other hand, in the present manuscript, T_{1spin} is much shorter than that (= more than 2 ms) in [20].

(2) In figure 5a, $T_{1orbital}$ is shown. I think authors should describe how to estimate it with information of the equipment to estimate the fast rates.

(3) In figure 2c, the intensity increases when ODMR occurs. Does it mean the fluorescence intensity increases when ODMR occurs? If so, does emission occur when ODMR occurs?

(4) I would like to ask how much the contrast of the ODMR signal. The "ODMR contrast" means how much the fluorescence intensity changes when ODMR occurs.

(5) Recently, researches on a single Germanium-vacancy related center in diamond has been reported. It has also excellent optical properties, which is similar with SiV. I think the authors should refer and introduce not only NV center but also them at the introduction part.

Minor comments

(1) The explanation about the "peak ratio" is not clear. Is it estimated by height or integral at some region? It may be better to write a figure to show the way how to estimate it.

(2) In line 126: It should be T_2^* , not T_2 .

Reviewer #3 (Remarks to the Author):

The manuscript "Coherent control of the silicon-vacancy spin in diamond" by Benjamin Pingault and coworkers was under review. It describes the coherent control of single, implanted silicon vacancy centers in diamond. Their main advantage against the commonly used nitrogen vacancy center is described as being based on the photo physical properties.

The work is novel and interesting. The paper is well written. Generally, the manuscript should be considered for publication.

A number of subtleties nevertheless limits the value of the presented manuscript:

All evidence on the presented 'spin control' relies on the 'peak-ratio' (line 74) - I believe that this is a limited measure, since the peak is limited in accuracy and likely dominated by experimental jitter (of the APDs and such). I believe that rather the peak area should be measured and compared.

Furthermore, no further error discussion, nor count rates, nor recording times are supplied with the entire analysis.

All images are of limited quality - especially since it would be easy possible to stuff them with valuable information for an experimental reproduction:

Fig. 1a: What are the splittings? e.g. between 1 + 3 and 2 + 4?

Fig. 1b: how long did this measurement take? The authors mean 'arb. units.' and not atomic units.

Fig. 1c: what is Δt ? (not indicated in 1b)

Fig. 2a: what is the timing? How long should the rf field be?

Fig. 2b: odd arrow heads

Fig. 2d: what is the gray dashed line? (not evident)

Fig. 3a: what is the exact timing? Why the delay of the rf pulse after the laser is turned off?

Fig. 3b: unhealthy inset into the figure. Should be rather on top of each other.

Fig. 4a: Why the delay of the rf pulse after the laser is turned off? What was the recording time?

b + c: not evident what the difference is between b and c (why not mark it into the figure?)

Fig. 5: please note that I am able to turn my head to the left OR to the right to read the frame label, but not to both sides.

In the section 'coherent control of the spin' the 'upward drift' is mentioned. In this context I ask

the authors to reconfirm that the presented data is raw data.

Minor details:

I wonder the angle in line 64 to be 109 degree. The tetraeder angle is 109.5 degree. Is this meant?

In conclusion, I recommend publication of the manuscript after the authors have addressed the above mentioned details. I underline that to enable other groups to follow the presented results should have been in the interest of the authors already. I wonder why none of the eight authors did see this limited reproducibility.

Reviewer #1 (Remarks to the Author):

The manuscript describes coherent microwave control of the electron spin of single SiV- in diamond. This topic is of interest because of the good optical quality of the fluorescence compared to the NV-. The results are novel, being an extension of the coherent population trapping work from 2014. The data quality is good and the presentation is very good (apart from the comments below). I recommend publication once the three questions below are resolved.

1. At the end of page 5 (line 123) measurements of $T_{1,orbital}$ are mentioned, but there is no mention in the manuscript of how these data were measured, or any presentation of these data. Please introduce these measurements in the text and show the data. If this refers to data from a previous paper then reference this clearly.

We thank the reviewer for this comment. In response, we have introduced the measurements of $T_{1,orbital}$ in the main text on page 6 (line 137): 'The orbital decay rate is obtained in the absence of magnetic field similarly to the spin population decay presented in Figs. 1b and c, with initialisation into the lower orbital branch of the ground state and measurement of the population recovery into the upper branch (see Supplementary Note 6)'. We have added a section to the supplementary materials (labelled section F) where we describe the experiment in more details and display example curves.

2. Why is the value of T_2^* longer than that in the 2014 CPT papers? Please explain this in the paper.

We have added the following explanation in the main text on page 5 (line 124): 'Shorter dephasing times have previously been obtained at slightly higher temperatures through coherent population trapping [19, 20], which provides a lower bound due to additional sources of dephasing, including noise from the driving fields. A strain-induced increase of orbital splitting to an energy at which phonon population is larger [25] is also a cause for shorter dephasing times.'

While during a Ramsey measurement, the spin is left to evolve freely between microwave pulses, a CPT experiment relies on constant driving of the system by two laser fields which induce dephasing through noise. In Rogers et al. (ref. 20 in the main text), an EOM is used to generate the second field from a laser. Although one can expect the noise from the sideband generation itself to be small, the authors do not comment on it. It is also worth noting that in their experiment, a helium flow cryostat was used to reach a temperature of about 5 K and the sample was mounted on a permanent magnet, which might not perfectly thermally connect the sample to the cold finger. Thus the temperature of the sample was most likely not as low as that in the present manuscript, leading to a shorter dephasing time (see Fig. 5 of the present manuscript). In Pingault et al. (ref. 19 in the main text), the investigated SiV was strained, probably due to the FIB milling to carve SILs, resulting in a larger orbital splitting energy of approximately 100 GHz at which phonon population is larger, hence a shorter coherence time.

3. The hyperfine is given as 70MHz, but in Rogers et al (PRL 113, 263602 (2014)) it is given as 35MHz. Presumably this is a different convention? Please explain in the paper.

In their article, Rogers et al. explain that the value of 35 MHz is a suggested level splitting based on the 69 MHz separation between the two measured CPT dips. In the present manuscript, we measure a splitting of about 54 MHz between the two ODMR peaks. From this splitting, we deduce a component of the hyperfine constant parallel to the SiV axis A_{\parallel} of 70 MHz through the model described in section D of the Supplementary materials. To clarify the reason for this difference in hyperfine level splitting but the consistency of the results, we have edited section D of the Supplementary materials to include on page 12 line 135 : 'The hyperfine level splitting results from an interplay between the hyperfine interaction, the electronic spin orbit coupling and the orientation and magnitude of the external magnetic field. In our experiment, the angle between the magnetic field and the symmetry axis of the SiV⁻ centre is approximately 109°, which leads to a 27 MHz splitting approximately, hence a 54 MHz splitting between the measured ODMR peaks. In ref. [3], this angle for the ²⁹SiV centre studied is not mentioned. Assuming an angle of 70° as for the ²⁸SiV measurement mentioned earlier in the text, we obtain a level splitting of approximately 35 MHz, consistent with the 69 MHz splitting between the two CPT dips observed experimentally.'

Further minor points:

1. It could be good to mention in the title that a single SiV⁻ is being used

We thank the reviewer for this suggestion. We have considered this and consulted other colleagues, and we have chosen to keep the original title as adding 'single' seems to suggest misleadingly that coherent control of ensembles of SiV centres has been reported.

2. On line 126 there is 1/T2 but presumably this should be 1/T2*

We thank the reviewer for noticing this typo. We have corrected it.

3. On lines 130, 136 and in the supplementary info there is 1/2T_{1,orbital} and 1/2T_{1,spin}. It would be clearer to put brackets around the denominator.

We have added the brackets.

4. The 2 panes of figure 5 are too close to each other so the axes labels are almost touching.

We have edited the figure.

Reviewer #2 (Remarks to the Author):

In this manuscript, the silicon-vacancy (SiV) center in diamond is investigated. The negatively charged SiV center emits around 80 % of its photons into the zero phonon line, with optical properties characterized by spectral stability and narrow inhomogeneous distribution in bulk diamond. In this manuscript, authors report the first realization of coherent control of the SiV center electron spin. They perform optically detected magnetic resonance (ODMR) by tuning the frequency of a microwave field into resonance with the Zeeman splitting between two electron spin states of a

single SiV center.

From the excellent optical properties, SiV center is significantly interesting. The electron spin states and its magnetic field dependences have been already reported. All optical initialization, readout, and coherent preparation of single SiV spin has also been reported. I agree that the observation by ODMR is important from the view point of coherent control of spin by microwave.

We thank the reviewer for this comment. We would like to point out that the previously reported coherent population trapping experiments lead to the preparation of a superposition of spin states which is unknown (no mention of any specific state being prepared in either of ref. 19 and 20). Furthermore, the prepared state changes from measurement to measurement based on the relative phases and powers of the two driving optical fields. Hence, such a preparation is of limited use for further spin control experiments. We here report the first coherent control of the SiV spin, where any predetermined spin state is reproducibly created. Such a control enables quantum computing algorithms, a simple example of which is the Ramsey sequence presented in this manuscript.

I have several questions and comments as follows.

(1) In previous study referred in [20], the coherence time (T_2^*) is reported to be 35 ns. I would like to ask the reason why $T_2^* = 115$ ns in the present study is much longer than it. The authors conclude the T_2^* is limited by $T_{1\text{spin}}$ in the present study. On the other hand, in the present manuscript, $T_{1\text{spin}}$ is much shorter than that (= more than 2 ms) in [20].

We have added the following explanation in the main text on page 5 (line 124): 'Shorter dephasing times have previously been obtained at slightly higher temperatures through coherent population trapping [19, 20], which provides a lower bound due to additional sources of dephasing, including noise from the driving fields. A strain-induced increase of orbital splitting to an energy at which phonon population is larger [25] is also a cause for shorter dephasing times.'

While during a Ramsey measurement, the spin is left to evolve freely between microwave pulses, a CPT experiment relies on constant driving of the system by two laser fields which induce dephasing through noise. In Rogers et al. (ref. 20 in the main text), an EOM is used to generate the second field from a laser. Although one can expect the noise from the sideband generation itself to be small, the authors do not comment on it. It is also worth noting that in their experiment, a helium flow cryostat was used to reach a temperature of about 5 K and the sample was mounted on a permanent magnet, which might not be ideal for the thermal connection of the sample to the cold finger. Thus the temperature of the sample was most likely not as low as that in the present manuscript, leading to a shorter dephasing time (see Fig. 5 of the present manuscript). In Pingault et al. (ref. 19 in the main text), the investigated SiV was strained, probably due to the FIB milling to carve SILs, resulting in a larger orbital splitting energy of approximately 100 GHz at which phonon population is larger, hence a shorter coherence time.

The spin dephasing time T_2^* is limited by the orbital population decay time $T_{1,\text{orbital}}$ as the primary source of dephasing (as shown in Fig. 5a), while the spin population decay time $T_{1,\text{spin}}$ is longer. Only once any other source of dephasing has been eliminated does the value of $2T_{1,\text{spin}}$ constitute a maximum boundary for T_2^* . This boundary has not been reached yet. In Rogers et al. (ref. 20 in the

manuscript), the authors show that $T_{1,\text{spin}}$ varies considerably with the orientation of the magnetic field with respect to the SiV symmetry axis, from a few tens of nanoseconds up to 2.4 ms (as reminded in on page 6 line 149 of the present manuscript). The orientation of the magnetic field in the present study is not aimed at maximising $T_{1,\text{spin}}$, as the focus is on ODMR, coherent control of spin and identification of the dominant spin dephasing mechanism. Optimisation of the magnetic field orientation will be an important step once T_2^* is only $T_{1,\text{spin}}$ -limited.

(2) In figure 5a, $T_{1,\text{orbital}}$ is shown. I think authors should describe how to estimate it with information of the equipment to estimate the fast rates.

We have introduced the measurements of $T_{1,\text{orbital}}$ in the main text on page 6 (line 137): 'The orbital decay rate is obtained in the absence of magnetic field similarly to the spin population decay presented in Fig. 1b and c, with initialisation into the lower orbital branch of the ground state and measurement of the population recovery into the upper branch (see Supplementary Note 6)'. We have added a section to the supplementary materials (labelled section F) where we describe the experiment in more details and display example curves. The equipment used is the same as for the other measurements and is described in the Methods section.

(3) In figure 2c, the intensity increases when ODMR occurs. Does it mean the fluorescence intensity increases when ODMR occurs? If so, does emission occur when ODMR occurs?

In the ODMR experiment, the spin is first initialised through optical pumping, leading to a reduction of fluorescence intensity (see Fig. 1b of the main text). When the microwave is resonant with one of the two electron spin-flipping transitions, it transfers population from the initialised spin state to the depleted spin state, leading to a partial recovery of the fluorescence when the optical read-out pulse is sent.

To clarify this point, we have edited the sentence starting on page 4 in line 83, which now reads: 'When on resonance with one of the two transitions, the microwave transfers population from the initialised spin state to the depleted one, resulting in a recovery of fluorescence during the read-out pulse and thus a peak in the ODMR spectrum ...'

(4) I would like to ask how much the contrast of the ODMR signal. The "ODMR contrast" means how much the fluorescence intensity changes when ODMR occurs.

We agree with the reviewer that the ODMR signal should be described in more details. We have thus added: '...with a contrast around 33%' on page 4 in line 85 of the main text, and: 'The contrast calculated as the ratio between the peak height and the baseline is about 33%.' in the caption of Fig. 2. We have furthermore added an entire section (labelled section C) to the Supplementary materials to discuss the ODMR signal.

(5) Recently, researches on a single Germanium-vacancy related center in diamond has been reported. It has also excellent optical properties, which is similar with SiV. I think the authors should refer and introduce not only NV center but also them at the introduction part.

We thank the reviewer for this comment. To our knowledge, two articles about fluorescence from Germanium-Vacancy centres in diamond (Palyanov et al. and Iwasaki et al.) have been published recently. However, the reported optical properties are comparable to other colour centres in diamond (see I. Aharonovich and E. Neu, *Adv. Opt. Mater.* 2, 911 (2014)) and do not match those of the Silicon-Vacancy centre (more emission into the phonon sideband with a Huang-Rhys factor of 0.5, compared to 0.3 for SiV, and a reported fluorescence of 200 kHz, compared to 6 MHz for SiV). Concurrently with the submission of the present manuscript, promising results have been reported on the Arxiv and we would be happy to cite them once they have been peer-reviewed. We have added a note at the end of the main text, which reads: '*Note added* – During the preparation of this manuscript, we became aware of related work on the germanium-vacancy centre [28].' Citing Siyushev, P. et al., Arxiv:1612.02947 (2016).

Minor comments

(1) The explanation about the “peak ratio” is not clear. Is it estimated by height or integral at some region? It may be better to write a figure to show the way how to estimate it.

We agree with the reviewer about the explanation of the peak ratio. We have thus edited Fig. 1b to show the integrated signal used for the calculation of the peak ratio, as well as the caption, which now includes: 'The dark grey areas under the leading edges of the initialisation and readout pulses correspond to a duration of 30 ns during which the signal is integrated to calculate the peak ratio.'. Finally, to avoid confusion, we have edited the main text on page 3 in line 73, which now reads: 'The spin state measurement corresponds to the peak ratio between the integrated areas under the leading edge of the readout pulse and that of the initialisation pulse.'

(2) In line 126: It should be T_2^* , not T_2 .

We have corrected this typo.

Reviewer #3 (Remarks to the Author):

The manuscript "Coherent control of the silicon-vacancy spin in diamond" by Benjamin Pingault and coworkers was under review. It describes the coherent control of single, implanted silicon vacancy centers in diamond. Their main advantage against the commonly used nitrogen vacancy center is described as being based on the photo physical properties.

The work is novel and interesting. The paper is well written. Generally, the manuscript should be considered for publication.

A number of subtleties nevertheless limits the value of the presented manuscript:

All evidence on the presented 'spin control' relies on the 'peak-ratio' (line 74) - I believe that this is a

limited measure, since the peak is limited in accuracy and likely dominated by experimental jitter (of the APDs and such). I believe that rather the peak area should be measured and compared.

We agree with the reviewer on this comment. The peak ratio was already calculated by integrating the signal during 30 ns under the leading edges of the initialisation and readout pulses. We acknowledge that this was not explained in the manuscript and we thus thank the reviewer for pointing out this oversight. To clarify this important point, we have edited Fig. 1b to show the integrated signal used for the calculation of the peak ratio, and we have edited the caption, which now includes: 'The dark grey areas under the leading edges of the initialisation and readout pulses correspond to a duration of 30 ns during which the signal is integrated to calculate the peak ratio.'. Finally, to avoid confusion, we have edited the main text on page 3 in line 73, which now reads: 'The spin state measurement corresponds to the peak ratio between the integrated areas under the leading edge of the readout pulse and that of the initialisation pulse.'

Furthermore, no further error discussion, nor count rates, nor recording times are supplied with the entire analysis.

We have edited the caption of Figs. 2c and d, which now include in line 317: 'separated by 53.7 ± 0.3 MHz (error from Lorentzian fit)' and in line 319: 'Error bars are smaller than dots'. We have also edited Fig. 3b to include error bars and have added the following to the caption: 'error bars correspond to the standard deviation of the peak ratio'. Finally, we have edited Fig. 4b and c to include error bars. For extra experimental parameters, we have added a section entitled 'Experimental details' in the Methods section and which mentions the repetition rates and integration times of the main curves of the manuscript. We have also extended the Supplementary section B, on page 10 line 88, which now includes: 'With this emitter, we routinely obtained count rates of approximately 20000 counts/s at saturation in continuous non-resonant excitation and of approximately 500 counts/s at saturation in continuous excitation resonantly with transition D1'.

All images are of limited quality - especially since it would be easy possible to stuff them with valuable information for an experimental reproduction:

Fig. 1a: What are the splittings? e.g. between 1 + 3 and 2 + 4?

The splitting between the two orbital branches of the ground state are mentioned in the main text on page 2 in line 36 ('orbital branches of the ground state, split by 50 GHz') and on page 6 in line 153 ('frequency around 50 GHz corresponding to the orbital splitting of the SiV'). The details of the splittings for this particular emitter are visible in Supplementary Fig. 3 and given in Supplementary section E on page 13 line 176:

$$\begin{aligned}E_4 &= 2\pi \cdot 51.605 \text{ GHz} \\E_5 &= 2\pi \cdot 51.630 \text{ GHz} \\E_6 &= 2\pi \cdot 54.237 \text{ GHz} \\E_7 &= 2\pi \cdot 54.261 \text{ GHz}\end{aligned}$$

We agree with the reviewer that this information can be included in Fig. 1a of the main text and have thus edited the figure to include the splitting labelled ΔE , with the addition to the caption: 'The ground state upper orbital branch lies approximately at $\Delta E = 50$ GHz above the lower one.'

Fig. 1b: how long did this measurement take? The authors mean 'arb. units.' and not atomic units.

We have corrected the labelling of the y-axis to 'arb. u.' to avoid confusion. The repetition rate and integration times for the measurement are now included in the section Experimental details of the Methods: 'The results presented in Fig. 1b have been acquired with a repetition rate of 500 kHz and an integration time of 300 s.'

Fig. 1c: what is Δt ? (not indicated in 1b)

We thank the reviewer for pointing out this mistake. We have modified Fig. 1b to include Δt .

Fig. 2a: what is the timing? How long should the rf field be?

We agree with the referee that those parameters should be mentioned in the manuscript. We have edited the x-axis of Fig. 1b and included in the caption that optical pulses are 500 ns long. For Fig. 2a, the caption now reads: 'A 140 ns microwave pulse is applied during the 160 ns interval between optical initialisation and readout pulses'. We have also added an entire section about the ODMR signal in Supplementary section C.

Fig. 2b: odd arrow heads

We thank the reviewer for this comment but think the shape of those arrow heads does not alter the understanding of this figure.

Fig. 2d: what is the gray dashed line? (not evident)

The grey dashed lines in Fig. 2d correspond to the transitions where both nuclear and electronic spins are flipped, as mentioned in the caption: 'the two overlapping dashed grey curves correspond to the transitions where both electronic and nuclear spins are flipped.'

Fig. 3a: what is the exact timing? Why the delay of the rf pulse after the laser is turned off?

The duration of the optical pulses remains unchanged from the previous figures and is mentioned in the caption of Fig. 1b to be 500 ns. The fixed delay between them is included in the caption: 'The time interval between initialisation and readout pulses is kept constant at 210 ns'. This delay is determined to accommodate the longest MW pulse duration and is kept constant even for shorter microwave pulses as we do not want to include the effect of spin T1 on top of the Rabi oscillations. To clarify this point, we have edited the caption: '...is kept constant at 210 ns, in order not to include the effect of the spin population decay into the measured signal'.

Fig. 3b: unhealthy inset into the figure. Should be rather on top of each other.

The inset presents data to confirm that coherent control has been achieved for both nuclear spin projections. We would thus prefer to keep this curve as an inset.

Fig. 4a: Why the delay of the rf pulse after the laser is turned off? What was the recording time?

This delay between the two optical pulses is determined to accommodate the longest MW pulse sequence and is kept constant even for shorter microwave pulse sequences as we do not want to include the effect of $T_{1,\text{spin}}$ on top of the Ramsey oscillations. To clarify this point, we have edited the caption which now reads: 'Microwave and optical pulses protocol: two microwave $\pi/2$ pulses of 8 ns each are separated by a variable delay τ . The time interval between the initialisation and readout pulses is held constant at 160 ns, in order not to include the effect of the spin population decay into the signal.' The recording time is now mentioned in the Methods, in section Experimental details: 'the results in Fig. 4b and c were measured with a repetition rate of 50 kHz (to minimise heating) with 2000 s integration and 200 kHz with 400 s integration respectively.'

b + c: not evident what the difference is between b and c (why not mark it into the figure?)

The difference between the two curves is mentioned in the caption and in the main text on page 5 in line 113: 'We fix the frequency of the microwave pulses such that in one case (Fig. 4b), the microwave frequency is in the middle of the two resonances, corresponding to a detuning of 27 MHz from each of them, and in the other case (Fig. 4c), the detuning from one resonance equals twice that from the other (36 MHz and 18 MHz, respectively). We agree with the reviewer that this difference can be indicated in the figure and have thus modified it accordingly.'

Fig. 5: please note that I am able to turn my head to the left OR to the right to read the frame label, but not to both sides.

The y-axes on the left side of both figures have to be the decoherence rates, instead of coherence times, to evidence the linear dependence of those rates with temperature. The y-axes on the right side of both figures correspond to the inverse of the one on the left, as indicated in the axis labels. This extra axis is only added for convenience for the reader who can then directly read the measured coherence times without having to calculate them based on the rates of the left y-axes. To avoid confusion, we have edited the caption which now mentions: 'In both figures, the y-axis on the right indicates the coherence times corresponding to the rates of the left axis.'

In the section 'coherent control of the spin' the 'upward drift' is mentioned. In this context I ask the authors to reconfirm that the presented data is raw data.

We confirm that this is raw data.

Minor details:

I wonder the angle in line 64 to be 109 degree. The tetraeder angle is 109.5 degree. Is this meant?

The reviewer is correct. This indication is due to the fact that we cannot guarantee that the angle is perfectly 109.5 degree. In order to avoid confusion, we have edited this value which now reads: '109.5° ± 1°'.

In conclusion, I recommend publication of the manuscript after the authors have addressed the above mentioned details. I underline that to enable other groups to follow the presented results should have been in the interest of the authors already. I wonder why none of the eight authors did see this limited reproducibility.

We are particularly grateful to the reviewer for pointing out this flaw of the previous version of the manuscript. We agree with him/her that this was necessary information in the interest of the reproducibility of those experiments. To us, this highlights the importance of the peer-reviewing process and we want to renew our thanks to the reviewer for this input.

REVIEWERS' COMMENTS:

Reviewer #1 (Remarks to the Author):

The three reviewers all recommended publication after response to many fairly minor points. These have all been dealt with to my satisfaction so I now recommend publication with no further changes.

Reviewer #2 (Remarks to the Author):

Dear Authors,

I think authors replied answers to all my questions.
As far as I see, the paper can be considered for publication.

Reviewer #3 (Remarks to the Author):

The revised version of "Coherent control of the silicon-vacancy spin in diamond" was under review. All my previous concerns are somehow addressed.

I have three remaining remarks:

- The splittings (e.g. $2 \pi \times 51.605$ GHz) show too many digits. I do not believe the authors, that they were able to determine this value down to a MHz! This should be corrected
- The odd arrow heads in Fig. 2b do indeed "not alter the understanding of the figure", but also do not help the reader to understand what's going on. My remark reflects mostly the overall limited quality of the figures. If the authors do not care, they should keep it.
- The same holds for Fig. 3b.

With this, I suggest again to carefully revise Fig. 2b and 3b. The manuscript is ready to be published.

REVIEWERS' COMMENTS:

Reviewer #1 (Remarks to the Author):

The three reviewers all recommended publication after response to many fairly minor points. These have all been dealt with to my satisfaction so I now recommend publication with no further changes.

Reviewer #2 (Remarks to the Author):

Dear Authors,

I think authors replied answers to all my questions.
As far as I see, the paper can be considered for publication.

Reviewer #3 (Remarks to the Author):

The revised version of "Coherent control of the silicon-vacancy spin in diamond" was under review. All my previous concerns are somehow addressed.

I have three remaining remarks:

- The splittings (e.g. 2π 51.605 GHz) show too many digits. I do not believe the authors, that they were able to determine this value down to a MHz! This should be corrected

We have removed the last digit on energies of the levels in the ground state upper branch extracted from the group theoretical model to reflect the fact that their uncertainties are of the order of 10 MHz.

The energies of the levels in the ground state lower branch have a much lower uncertainty of 0.2 MHz, being obtained from ODMR scans.

- The odd arrow heads in Fig. 2b do indeed "not alter the understanding of the figure", but also do not help the reader to understand what's going on. My remark reflects mostly the overall limited quality of the figures. If the authors do not care, they should keep it.

We have corrected the arrow heads.

- The same holds for Fig. 3b.

We have corrected the appearance of the figure following the recommendation of the reviewer to plot curves on top of each other.

With this, I suggest again to carefully revise Fig. 2b and 3b. The manuscript is ready to be published.